# RETHINKING MISSING MODALITY LEARNING: FROM A DECODING VIEW

## ABSTRACT

Conventional pipeline of multimodal learning consists of three stages, including encoding, fusion, and decoding. Most existing methods under missing modality condition focus on the first stage and aim to learn the modality invariant representation or reconstruct missing features. However, these methods rely on strong assumptions (i.e., all the pre-defined modalities are available for each input sample during training and the number of modalities is fixed). To solve this problem, we propose a simple yet effective method called Interaction Augmented Prototype Decomposition (IPD) for a more general setting, where the number of modalities is arbitrary and there are various incomplete modality conditions happening in both training and inference phases, even there are unseen testing conditions. Different from the previous methods, we improve the decoding stage. Concretely, IPD jointly learns the common and modality-specific task prototypes. Considering that the number of missing modality conditions scales exponentially with the number of modalities $O(2^n)$ and different conditions may have implicit interaction, the low-rank partial prototype decomposition with enough theoretical analysis is employed for modality-specific components to reduce the complexity. The decomposition also can promote unseen generalization with the modality factors of existing conditions. To simulate the low-rank setup, we further constrain the explicit interaction of specific modality conditions by employing disentangled contrastive constraints. Extensive results on the newly-created benchmarks of multiple tasks illustrate the effectiveness of our proposed model.

## 1 INTRODUCTION

Multimodal learning is recently one of the increasingly popular yet challenging tasks involved in both computer vision and natural language processing Ben-Younes et al. (2017); Do et al. (2019); Gabeur et al. (2020); Liu et al. (2018). The target of multimodal learning is to utilize complementary information contained in multimodal data for improving the performance of various tasks. Many superior approaches on multimodal learning have been well developed in an ideal situation. However, a common assumption underlying these approaches is the completeness of modality (i.e., the full modalities are available in both training and testing data). In practice, such an assumption may not always hold in real world due to some overwhelming reasons. For example, some uploaded YouTube videos do not have audio tracks, also, the black screen may occur during the broadcast, leading to the lack of visual modality.

Although a bunch of endeavors are devoted to developing effective methods to cope with the missing modality conditions in the training and inference stages, there is no unified paradigm that takes into account all possible scenarios. For instance, Pham et al. (2019); Zhao et al. (2021) only consider the incompleteness of testing data, however, obtaining a lot of complete data for training is extremely labor-intensive. Ma et al. (2021) formulates a new setting that considers the incompleteness of training data, while there are only two modalities used. As we know, due to the quick developments of feature-extraction skills, one data sample may have more than two kinds of modality representation.

In this paper, we focus on a more general setting (as shown in Fig. 1), where the number of modalities is arbitrary and there are various incomplete modality conditions, to systematically study the missing modality problem. We propose a simple yet effective method called Interaction Augmented Prototype Decomposition (IPD) to capture the universality and particularity of different modality conditions

(including both complete and incomplete conditions). To the best of our knowledge, we try to study generalizable missing modality learning from the decoding perspective for the first time. Following Zhou et al. (2021; 2022), we treat the weights of the standard linear classifier as task prototypes. Since different modality conditions correspond to different task prototypes, training multiple models separately for all the conditions is time-consuming, IPD jointly learns the common and modality-specific prototypes. Considering that the number of missing modality conditions scales exponentially with the number ($n$) of modalities $\mathbf{O}(2^n)$ and different conditions may have implicit interaction, the low-rank partial prototype decomposition with enough theoretical analysis is employed for modality-specific component to reduce the complexity. To simulate the low-rank setup, we further constrain the explicit interaction of specific modality conditions by employing disentangled contrastive constraints. We conduct extensive experiments on the newly-created benchmarks of multiple tasks, the experimental results show that IPD could achieve competitive results compared with the state-of-the-art methods of conventional multimodal learning. The main contributions can be summarized as follows:

- We propose a novel method called Interaction Augmented Prototype Decomposition (IPD) for generalizable missing modality learning, which jointly learns the common and modality specific prototypes. To the best of our knowledge, it is the first time to improve the decoding stage.

- Considering that the number of missing modality conditions scales exponentially with the number ($n$) of modalities $\mathbf{O}(2^n)$ and different conditions may have implicit interaction, the low-rank partial prototype decomposition with is employed to reduce the complexity. The decomposition also can promote unseen generalization with the modality factors of existing conditions.

- We constrain the explicit interaction of specific modality conditions by employing disentangled contrastive constraints.

- We conduct low-rank ensemble learning to enhance the performances of the conditions with relatively few modalities.

## 2 RELATED WORK

### 2.1 CONVENTIONAL MULTIMODAL LEARNING

Multimodal learning utilizes complementary information contained in multimodal data to improve the performance of various tasks. The key point of this area is multimodal fusion. Early fusion methods are mainstream and integrate features of different modalities before feeding them to the task modules. For example, concatenating different features Zadeh et al. (2016b) is a simple way. Zadeh et al. (2017) proposes a product operation to allow more interaction among different modalities during the fusion process. Liu et al. (2018) considers the large complexity of Zadeh et al. (2017) and utilizes modality-specific factors to achieve efficient low-rank fusion. However, the intra-modal dynamics cannot be effectively captured in the above methods. Liang et al. (2019) employs low-rank fusion for each time step of multi-view sequential input. Tsai et al. (2019) utilizes Transformer to replace LSTM due to the powerful encoding capacity. Rahman et al. (2020) employs large-scale pre-trained Bert embeddings for textual modeling. Liang et al. (2021) proposes modality-invariant crossmodal attention towards learning crossmodal interactions over modality-invariant space in which the distribution mismatch between different modalities is well bridged. However, these methods do not consider the missing modality problems at all. For this aim, we propose IPD to solve the problems.

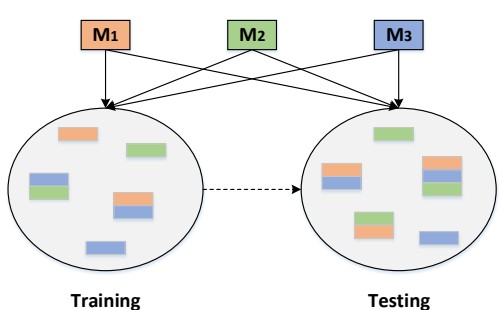

Figure 1: An example of generalizable missing modality learning, where $M_1, M_2, M_3$ denote three modalities. There are **unseen modality combinations** in the testing stage.

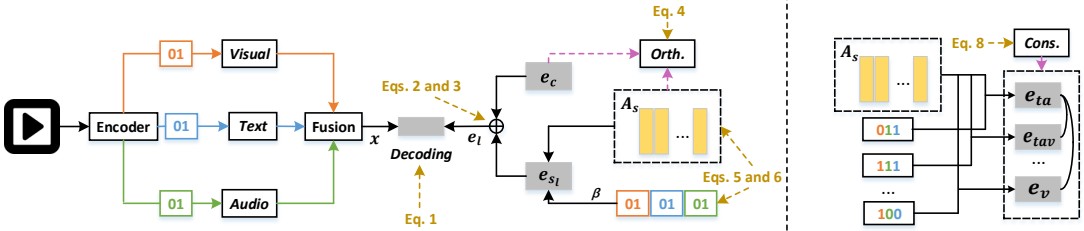

Figure 2: The framework of IPD, where the left part denotes the low-rank partial decomposition and the right part denotes the disentangled contrastive constraints. Different color corresponds to different modalities (red denotes visual modality, blue denotes text modality, green denotes audio modality). "0" and "1" denote the states of existence and nonexistence. In the left part, the process of Eq. 1 to Eq. 6 is included. "Orth." denotes orthogonal constraints (Eq. 4). Yellow bars denote $A_s$ that consists of $a_{s_j}$. Gray bars of $e_c$ and $e_{s_l}$ denote the common and modality-specific prototypes. $e_{s_l}$ can be calculated with $A_s$ and $\beta$ through Eqs. 2 and 3. The decoding process (Eq. 1) is developed by correlating $x$ and $e_l$. In the right part, we mainly present Eq. 8 of contrastive constraints. "Cons" denotes contrastive learning (Eq. 8). The gray bars of $e_{ta}, e_{tav}, e_v$ denote the modality specific prototypes of $ta, tav, v$.

## 2.2 MULTIMODAL LEARNING WITH INCOMPLETENESS

Multimodal learning with incompleteness is another topic of concern. There are also two lines of incomplete multimodal learning. One is to process the representation noise, for example, the visual features of several time steps are not available, while all the remaining time steps are normal. Liang et al. (2019) proposes rank constraints to solve this problem. The other is to process the missing modality conditions that are more likely to be encountered in real-world applications. Also, there exist more researches that study it. Parthasarathy & Sundaram (2020) proposes a strategy to randomly discard the visual input during training at the clip or frame level to mimic real-world missing modality scenarios for audio-visual multimodal emotion recognition. Tran et al. (2017); Zhao et al. (2021) proposes Cascaded Residual Autoencoder (CRA) to utilize the residual mechanism over the autoencoder structure, which can take the corrupted data and estimate a function to well restore the incomplete data. Pham et al. (2019) proposes a sequential translation-based model to learn the joint representation between the source modality and multiple target modalities. Although many efforts are devoted to developing effective methods to cope with the missing modality conditions during training and inference, there is no unified paradigm considering all possible scenarios. Thus, we focus on a more general setting. Also, inspired by the usage of low-rank decomposition in multi-view interaction Liu et al. (2018); Do et al. (2019); Piratla et al. (2020), we introduce it into the task.

## 3 APPROACH

### 3.1 PROBLEM FORMULATION

We formulate the problem setting for generalizable missing modality learning in this section. Suppose the number of modalities is $n$, there are various modality scenarios in both training and inference stages. Concretely, about $2^n - 1$ scenarios [1] are available, since each modality has two states (i.e. 0-nonexistence, 1-existence). Under such real-world conditions, there are two challenging problems, one is to obtain good performances of existing modality combinations. The other is to generalize to unseen conditions, when the training set does not cover all the modality scenarios.

### 3.2 COMMON-SPECIFIC JOINT LEARNING

Following the previous work, we employ different encoding blocks for corresponding tasks. For multimodal sentiment analysis and speaker traits recognition, we adopt the same structure as LMF Liu et al. (2018), where LSTMs are utilized to process the multimodal features. As for multimodal

---

[1] For convenience, we utilize $2^n$ in the following sections.

video retrieval, we borrow the idea from MMT Gabeur et al. (2020), where the multimodal features are encoded by Transformer. To fuse the multimodal features, we use the mainstream tensor-based method, LMF, which considers the fine-grained interaction among different modalities. The training paradigms of multimodal sentiment analysis and speaker traits recognition are similar to the settings of LMF. As for the training objective of video retrieval, we employ the contrastive cross-entropy loss and transform it to a classification task (i.e., the matched pairs are positive samples)

We mainly focus on classification and regression. Similar to Zhou et al. (2021), Zhou et al. (2022), we treat the linear weights of the final layer as task prototypes of different modality scenarios. There are $2^n$ scenarios and the examples $(x, y)$ ($x \in \mathbb{R}^m$ denotes the fusion result of multiple modalities, $y \in \mathbb{R}$ denotes the label or regression score) from specific scenario $l \in [2^n]$ are generated as:

$$y = x^\top e_l, \quad e_l = e_c + e_{s_l}, \tag{1}$$

where $e_c \in \mathbb{R}^m$ denotes the common prototypes of different modality combinations, $e_{s_l} \perp e_c \in \mathbb{R}^m$ denotes the modality-specific prototypes of $l$-th modality scenarios, $e_l$ denotes the complete task prototype for $l$-th modality combination. Intuitively, the common component contains more high-level semantic information (i.e., the reasoning of performance), and the modality-specific part contains more low-level detailed information (i.e., the loudness of voice, the movement range). $y = \pm 1$ for classification tasks and $y \in [-p, p]$ for regression tasks.

## 3.3 LOW-RANK PARTIAL PROTOTYPE DECOMPOSTION

Based on Eq. 1, there exist superior scenario specific classifiers $\{e_l | l \in [2^n]\}$, one for each modality scenario, such that $e_l = e_c + e_{s_l}$. Note that all these classifiers share the common component $e_c$. If we are able to find multiple specific classifiers of this form, $e_c$ and $e_{s_l}$ could be extracted from them. Although the prototypes cover all the modality combinations, the complexity ($O(2^n)$) would be large when $n$ increases. Further, training the modality-specific prototypes separately is inefficient with neglecting the interaction among different modality scenarios. Therefore, we reformulate the $E_s \in \mathbb{R}^{m \times 2^n}$ that consists of $e_{s_i}$ as $A_s \cdot \beta^\top$, where $A_s \in \mathbb{R}^{m \times k}$, $\beta \in \mathbb{R}^{2^n \times k}$, and $k$ denotes the low-rank value. Our idea can be extended from Eq. 1:

$$y = x^\top \left(e_c + \sum_{j=1}^{k} \beta_{l,j} a_{s_j}\right) = x^\top (e_c + A_s \beta_l^\top) \tag{2}$$

where $a_{s_j} \perp e_c \in \mathbb{R}^m$, and $a_{s_j} \perp a_{s_q}$ for $j, q \in [k], j \neq q$ are modality specific features. The correlation between the $a_{s_j}$ and task is given by the coefficients $\beta_l \in \mathbb{R}^k$, varies among multiple modality combinations. Under this setting, there also exists superior modality specific classifier $e_l$ such that $e_l = e_c + A_s \beta_l^\top \in \mathbb{R}^m$, where $e_c \in \mathbb{R}^m, A_s \in \mathbb{R}^{m \times k}, \beta_l \in \mathbb{R}^k$ are trainable variables. More concretely, the trainable task prototypes can be written as:

$$E = e_c \mathbf{1}^\top + A_s \beta^\top = [e_c, A_s][\mathbf{1}, \beta]^\top \tag{3}$$

where $E = [e_1, e_2, ..., e_{2^n}] \in \mathbb{R}^{m \times 2^n}$, $\mathbf{1} \in \mathbb{R}^{2^n}$ denotes the all ones vector and $\beta^\top = [\beta_1, \beta_2, ..., \beta_{2^n}] \in \mathbb{R}^{k \times 2^n}$. However, in practice, given a general matrix $E$ which can be written as $E = e_c \mathbf{1}^\top + A_s \beta^\top$, there are multiple ways of decomposing $E$ into this form, $e_c$ and $A_s \beta^\top$ cannot be uniquely determined by the decomposition alone. Therefore, we conduct constraints to the $[e_c, A_s]$ with orthogonal regularization:

$$\mathcal{L}_o = ||I_{k+1} - [e_c, A_s]^\top [e_c, A_s]||^2 \tag{4}$$

where $\mathcal{L}_o$ denotes the regularization loss, $I_{k+1} \in \mathbb{R}^{(k+1) \times (k+1)}$ denotes identity matrix. Such regularization term caters the following propositions:

**Proposition 1:** When $e_c \perp \text{Span}(A_s)$, the decomposition of $E = e_c \mathbf{1}^\top + A_s \beta^\top$ has a unique solution. Suppose $E = e_c \mathbf{1}^\top + A_s \beta^\top = w_c \mathbf{1}^\top + W_s \Gamma^\top$ is a rank-$(k+1)$ matrix, where $A_s \in \mathbb{R}^{m \times k}$,

$\beta \in \mathbb{R}^{2^n \times k}$, $W_s \in \mathbb{R}^{m \times k}$, and $\Gamma \in \mathbb{R}^{2^n \times k}$ are all rank-$k$ matrices with $k \leq \min(m, 2^n)$. When $e_c \perp \text{Span}(A_s)$, $w_c = e_c$ is equal to $w_c \perp \text{Span}(W_s)$.

**Proposition 2:** If the orthogonal regularization is not satisfied ($e_c$ is not orthogonal to $\text{Span}(A_s)$), the performance of partial modality combinations will be influenced.

### 3.4 HIERARCHICAL DECOMPOSITION OF $A_s \beta^\top$ AND UNSEEN GENERALIZATION

Although the decomposition of $A_s \beta^\top$ results in the change of complexity (from $m \cdot 2^n$ to $k \cdot 2^n$), the exponential term $2^n$ still exits. Furthermore, the decomposition of $A_s \beta^\top$ does not consider the fine-grained interaction among different modality combinations. To solve this problem, we employ the hierarchical decomposition for the high-order tensor $\beta \in \mathbb{R}^{2^n \times k}$. Concretely, we divide $\beta$ into a series of $n$-order tensors:

$$\beta = [C^1, C^2, ..., C^k] \tag{5}$$

where $C^j \in \mathbb{R}^{2^n}$, we apply CP decomposition Harshman et al. (1970) to each $C^j$:

$$C^j = \sum_{r=1}^{R} \bigotimes_{i=1}^{n} C_{i,r}^j, \quad C_{i,r}^j \in \mathbb{R}^2 \tag{6}$$

where $\otimes$ denotes tensor outer product operation over a set of vectors, $R$ denotes the rank value of CP decomposition. Such operations change the complexity of $\beta$ from exponential to linear. With the approximation, the weights for unseen modality combinations can also been obtained with tensor multiplication. We employ these weights for the evaluation of missing modality generalization.

### 3.5 DISENTANGLED CONTRASTIVE CONSTRAINTS FOR DECOMPOSITION

The rank value is related to the interaction among different modality combinations to some extent. To further enhance the explicit interaction of specific modality combinations and control the rank value, we employ disentangled contrastive constraints for decomposition. Specifically, we suggest employing the task prototype (i.e. $e_{s_i}$) of the modality-specific part. Different from the obscure modality gaps that cannot be obtained, the interaction between different modality combinations is obvious. For example, suppose that there are totally three modalities $(t, a, v)$ and the possible combinations include $\{t, a, v, ta, tv, av, tav\}$. We can easily find that $ta$ is more related to $t$, $a$, and $tav$ than $v$, since $ta$ and $v$ do not have intersection. We express the above statements as:

$$D(ta, \eta) \geq D(ta, v), \quad \eta \in \{t, a, tav\} \tag{7}$$

where $D()$ denotes the correlation function. Following the laws of nature, we conduct disentangled contrastive constraints to the modality-specific part. The naturally similar pairs like $ta$ and $tav$ should have higher matching scores than negative pairs like $ta$ and $v$,

$$\mathcal{L}_c = \max\Big(0, \Delta - h(e_{ta}, e_{tav}) + h(e_{ta}, e_v)\Big) \tag{8}$$

where $e_{ta}, e_{tav}, e_v$ denote the modality-specific prototypes, $\Delta$ is the margin value, $h()$ denotes the normalized inner product operation (cosine similarity). In practice, the number of existing positive and negative pairs also scales exponentially with the number of modalities, therefore, we utilize the maximum suppression sampling scheme to reduce it. Concretely, for a specific input (i.e. $e_{ta}$), we consider the combination (i.e. $e_{tav}$) with most modalities that contain the processed modalities as a positive object, the complementary set (i.e. $e_v$) is treated as a negative object.

### 3.6 LOW-RANK ENSEMBLE LEARNING

With the approximation of $\beta$, we employ low-rank ensemble learning as an auxiliary, since the samples with more modalities could enhance the training of more modality factors produced by Eq.

6 (e.g. a sample with modalities $ta$ can be used for training $e_{ta}, e_t, e_a$ simultaneously). Concretely, a data point with $n^*$ modalities can be augmented to $2^{n^*}$ modality combinations. We calculate the average results of all the augmented modality scenarios of an input sample. Suppose that the encoded features are represented as $u_1, u_2, ...u_n \in \mathbb{R}^m$, where $u_i$ denotes the feature of $i$-th modality. If a modality is missing, the original input would be replaced with all zeros vector, the encoded features are represented as $o_1, o_2, ...o_n \in \mathbb{R}^m$. Thus, if the $i$-th modality is missing, $u_i = o_i$. We concatenate each $W_i^b u_i$ and corresponding $W_i^b o_i$ to $v_i = [W_i^b u_i, W_i^b o_i] \in \mathbb{R}^{m \times 2}$, where $W_i^b \in \mathbb{R}^{m \times m}$ denotes the linear mapping weights following LMF. The fusion results for all the modality combinations can be denoted as $V \in \mathbb{R}^{m \times 2^n}$, where $V_{t_1, t_2, ..., t_n} = \prod_{i=1}^n v_{i, t_i} \in \mathbb{R}^m$. The result $O_e = O_c + O_s$ consisting of common $O_c$ and specific $O_s$ parts can be calculated as follows:

$$
O_c = \text{sum}\Big(\Big[\prod_{i=1}^n v_i \cdot \mathbf{1}\Big] \odot e_c\Big)/M
$$
$$
O_s = \text{sum}\Big(\Big[\sum_{r=1}^R \prod_{i=1}^n v_i C_{i,r}\Big] \odot A_s\Big)/M
$$
(9)

where $\odot$ denotes element-wise multiplication, sum() denotes summation operation for all the elements of the vector, $C_{i,r} \in \mathbb{R}^{2 \times k}$, $M$ denotes the ensembled number (i.e. $2^{n^*}$) which is related to $n^*$. The detailed derivation is shown in the appendix (section A).

## 4 TRAINING AND INFERENCE

The overall framework is shown in Fig. 2. We employ $\mathcal{L}_t = f(y^*, y)$ to denote the prediction loss, where $y^*$ denotes the prediction result with LMF and $y$ is the ground-truth label, $f()$ can be MAE for sentiment analysis or cross-entropy loss function for retrieval. $\mathcal{L}_e = f(O_e, y)$ denotes the loss when using ensemble learning. The final optimization objective is $\mathcal{L}_t + \lambda_1 \mathcal{L}_e + \lambda_2 \mathcal{L}_o + \lambda_3 \mathcal{L}_c$.

During inference, we employ $E = [e_1, e_2, ..., e_{2^n}]$ for both existing modality combinations and unseen ones. The unseen task prototypes can be calculated by the trainable modality factors as introduced in Eq. 6. We provide the training and inference details in the appendix (Algs. 1 and 2).

## 5 EXPERIMENTS

### 5.1 DATASET AND METRICS

We evaluate our method on three challenging tasks, multimodal sentiment analysis, multimodal speaker traits recognition, and multimodal video retrieval. In this section, we provide a brief introduction of the datasets and metrics:

**CMU-MOSI** Zadeh et al. (2016a): It is a collection of 93 opinion videos from YouTube movie reviews. Each video consists of multiple opinion segments (2199 segments in total) and each segment is annotated with the score in the range $[-3, 3]$, where $-3$ and $3$ indicate highly negative and positive. We report the metrics of BA (binary accuracy), F1, Corr (Correlation Coefficient), MA (Multi-class accuracy, higher is better), MAE (Mean-Absolute Error, lower is better).

**POM** Pérez-Rosas et al. (2013): POM is a speaker traits recognition dataset made up of 903 movie review videos. Each video has 16 speaker traits. We report the multi-class accuracy of different traits.

**MSR-VTT** Xu et al. (2016): MSR-VTT is composed of 10K YouTube videos, collected using 257 queries from a commercial video search engine. Each video is 10 to 30s long, and is paired with 20 natural sentences describing it. We report the common metrics of R@K and MdR.

### 5.2 DATA PREPROCESSING

**CMU-MOSI and POM.** Each dataset (CMU-MOSI, POM) consists of three modalities, including textual, visual, and audio modalities. For textual features, we employ the pre-trained 300-dimensional Glove embeddings Pennington et al. (2014). For visual features, we utilize Facet iMotions (2017)

Table 1: The results on CMU-MOSI.

| Method | Existing Combs. | | | | | Unseen Combs. | | | | |
|--------|------|------|------|------|------|------|------|------|------|------|
| | BA | F1 | MAE | Corr | MA | BA | F1 | MAE | Corr | MA |
| Random Drop | 61.7 | 61.8 | 1.594 | 0.277 | 15.8 | 70.6 | 70.5 | 1.252 | 0.482 | 26.4 |
| TFN Zadeh et al. (2017) | 62.7 | 62.2 | 1.407 | 0.317 | 19.8 | 71.6 | 71.1 | 1.090 | 0.513 | 28.6 |
| MFN Zadeh et al. (2018a) | 64.3 | 64.1 | 1.329 | 0.381 | 24.0 | 73.2 | 73.0 | 1.012 | 0.577 | 32.9 |
| LMF Liu et al. (2018) | 63.3 | 63.6 | 1.276 | 0.417 | 22.7 | 72.2 | 72.5 | 1.037 | 0.613 | 31.6 |
| T2FN Liang et al. (2019) | 64.3 | 64.1 | 1.250 | 0.428 | 23.0 | 72.7 | 72.8 | 1.055 | 0.610 | 31.9 |
| MulT Tsai et al. (2019) | 64.8 | 64.7 | 1.236 | 0.439 | 24.5 | 73.5 | 73.6 | 1.015 | 0.620 | 33.2 |
| MVAE Wu et al. (2018) | 65.1 | 65.3 | 1.283 | 0.425 | 25.0 | 74.9 | 74.7 | 0.997 | 0.608 | 32.5 |
| MCTN Pham et al. (2019) | 64.8 | 65.1 | 1.244 | 0.456 | 24.7 | 73.7 | 74.0 | 1.007 | 0.607 | 32.5 |
| MMIN Zhao et al. (2021) | 65.4 | 65.6 | 1.208 | 0.445 | 25.6 | 74.5 | 74.0 | 1.054 | 0.618 | 31.9 |
| **IPD (Ours)** | **67.7** | **68.0** | **1.142** | **0.486** | **27.9** | **76.6** | **77.4** | **0.984** | **0.625** | **34.0** |

Table 2: Ablation study on CMU-MOSI.

| Method | Existing Combs. | | | | | Unseen Combs. | | | | |
|--------|------|------|------|------|------|------|------|------|------|------|
| | BA | F1 | MAE | Corr | MA | BA | F1 | MAE | Corr | MA |
| w/o. All | 63.3 | 63.6 | 1.276 | 0.417 | 22.7 | 72.2 | 72.5 | 1.037 | 0.613 | 31.6 |
| w/o. LR | 66.7 | 67.0 | 1.180 | 0.459 | 26.2 | 73.8 | 73.6 | 0.998 | 0.617 | 32.8 |
| w/o. Orth | 65.5 | 65.6 | 1.207 | 0.449 | 24.2 | 73.6 | 73.7 | 1.007 | 0.622 | 32.4 |
| w/o. Ens | 67.5 | 67.7 | 1.161 | 0.464 | 26.8 | 75.2 | 75.3 | 0.994 | **0.627** | 33.0 |
| w/o. Cs | 67.0 | 66.7 | 1.195 | 0.467 | 26.4 | 74.5 | 74.3 | 0.987 | 0.623 | 33.2 |
| IPD (Full) | **67.7** | **68.0** | **1.142** | **0.486** | **27.9** | **76.6** | **77.4** | **0.984** | 0.625 | **34.0** |

to indicate 35 facial action units, which records facial muscle movement for representing the basic and advanced emotions. For audio features, we use COVAREP Degottex et al. (2014) acoustic analysis framework. To align the different modalities along the temporal dimension, we perform word alignment with P2FA Yuan & Liberman (2008).

**MSR-VTT.** The videos contain abundant multimodal information. Thus, we use multiple pre-trained models for extracting features. Concretely, we utilize seven modality experts: motion, scene, OCR, audio, speech, face, appearance following Gabeur et al. (2020).

## 5.3 MISSING MODALITY SETTING

We evaluate IPD from two views. One is the performances of existing modality combinations, the other is the performances of unseen combinations. Concretely, we divide CMU-MOSI, POM into $2^3 - 1 = 7$ pieces (scenarios). One piece is kept for the evaluation of unseen generalization (Since the combinations of fewer modalities can be augmented with those of more modalities, we employ the complete modality combination for unseen evaluations). As for the remaining 6 pieces, We employ 70%, 10%, 20% of the samples for training, validation, and testing for existing modality combinations[2]. To be realistic, we randomly generate the ratio of 7 pieces (by giving each piece a number from 0 to 1 and employing normalization) for 5 times and report the average results.

For the simulation of a large number of the modality combinations, we conduct video retrieval experiments on MSR-VTT, which contains rich multimodal information. Since the number of modalities is 7, there are $2^7 - 1 = 127$ combinations. For the evaluation of unseen modality combinations, we employ 8 pieces (the combinations with more modalities). The remaining pieces are divided in the proportion of previous work Gabeur et al. (2020) (90% for training, 10% for testing). We follow CMU-MOSI to randomly generate ratio and report the average results.

---

[2]The experimental details of all the tasks are shown in the appendix.

Table 3: The performances on POM, where we report the multi-class accuracy of four traits , including Credible (Cre), Vivid (Viv), Expertise (Exp), Entertaining (Ent).

| Method | Existing Combs. | | | | Unseen Combs. | | | |
|---|---|---|---|---|---|---|---|---|
| | Cre | Viv | Exp | Ent | Cre | Viv | Exp | Ent |
| Random Drop | 19.6 | 23.2 | 19.7 | 24.5 | 25.6 | 27.6 | 25.6 | 29.6 |
| TFN Zadeh et al. (2017) | 23.2 | 23.2 | 19.7 | 24.5 | 28.1 | 27.6 | 25.6 | 29.6 |
| MFN Zadeh et al. (2018a) | 19.6 | 23.6 | 20.7 | 26.1 | 25.6 | 29.6 | 26.6 | 31.0 |
| LMF Liu et al. (2018) | 19.6 | 23.2 | 20.7 | 28.1 | 25.6 | 27.6 | 26.6 | 33.5 |
| T2FN Liang et al. (2019) | 23.2 | 23.2 | 22.7 | 28.1 | 29.6 | 27.6 | 26.6 | 33.5 |
| MulT (Tsai et al., 2019) | 23.2 | 23.6 | 22.7 | 28.1 | 28.1 | 29.6 | 27.6 | 33.5 |
| MVAE Wu et al. (2018) | 23.2 | 23.6 | 25.1 | 26.1 | 29.6 | 29.6 | 27.6 | 31.0 |
| MCTN Pham et al. (2019) | 23.2 | 23.2 | 20.7 | 24.5 | 29.6 | 27.6 | 25.6 | 31.5 |
| MMIN Zhao et al. (2021) | 25.6 | 27.6 | 25.1 | 28.1 | 30.5 | 30.5 | 27.6 | **33.5** |
| IPD (Ours) | **27.1** | **30.5** | **27.1** | **30.5** | **34.5** | **36.9** | **31.0** | 33.0 |

## 5.4 EXPERIMENTS FOR MULTIMODAL SENTIMENT ANALYSIS

**Results:** Table 1 presents the overall comparison of IPD and existing methods on CMU-MOSI (both existing and unseen combinations). Note that we mainly compare IPD with the methods that use the same features for fairness. As for the evaluation of existing modality combinations, we could observe that MFN, TFN, LMF, and MulT perform worse than IPD as they pay more attention to the usage of complete modalities, leading to the poor adaptation on the missing modality scenarios. Even MulT utilizes Transformer Vaswani et al. (2017). Besides, IPD achieves the best performances on all the metrics among the existing missing-modality learning methods (including MVAE, MCTN, MMIN). Particularly, the performance of IPD increases the "MA" from $25.6$ to $27.9$ compared to the best counterparts. We carefully analyze the observations: (1) The mainstream reconstruction based methods (i.e. MCTN, MMIN) depends on the existence of all the modalities to obtain the supervised information in the training stage, therefore, when the samples for training are imcomplete, there will be a big drop in performance. (2) The methods about modality invariant representation (i.e. MVAE) are also widely studied, however, since the noise covariance matrix in Eq. 1 varies across different modality combinations and samples, none of the features have the same distribution. Further, the assumptions for training of MVAE also include the availability of all of the pre-defined modalities. As for the evaluation of unseen modality combinations, IPD performs better than all the baseline methods, which demonstrates that IPD has competitive generalization ability. In general, the best performances of IPD attribute to the advanced prototype decomposition which captures modality-shared knowledge and modality-specific knowledge, respectively, as well as the disentangled contrastive constraints that enhances the interaction of different modality conditions.

**Ablation Study:** We set some control experiments on CMU-MOSI to verify the effectiveness of IPD and the results are shown in Table 2. "w/o. All" denotes the model without all the contributions and equals to LMF. "w/o. LR" denotes that all the orthogonal task prototypes are trained separately without other contributions (i.e. low-rank approximation, contrastive constraints, augmented ensemble). "w/o Orth" denotes that we just remove the orthogonal constraints for $e_c$ and $A_s$ from the complete model. "w/o. Ens" denotes that we just remove the low-rank ensemble learning from the complete model. "w/o. Cs" denotes the model without contrastive constraints for the modality specific prototypes. From the Table 2, we could find that "w/o. All" achieves worst results on all the metrics, which is consistent with the objective law, since all the contributions for missing-modality learning are removed. The results of "w/o. LR" is similar to those of "IPD (Full)", we analyze that the model without low-rank constraints can also learn limited multimodal interaction from scratch. "w/o. Orth" achieves limited improvement based on the baseline method, which fits Proposition 2, the performance of some modality combinations would be influenced. The bad performances of "w/o. Ens" reveal the effectiveness of data augmentation. "w/o. Cs" achieves relatively bad results compared with IPD (Full), since the interaction enhancement among different modality combinations is important to simulate the low-rank condition.

We utilize low-rank approximation for $E_s$ and $\beta$, the rank values $k, R$ should be considered. We examine the performances of IPD on CMU-MOSI (Existing Combs.) with different values of $k, R$,

Table 4: Retrieval performances (text-to-video) on the MSR-VTT dataset.

| Method | Existing Combs. | | | | Unseen Combs. | | | |
|---|---|---|---|---|---|---|---|---|
| | R@1↑ | R@5↑ | R@10↑ | MdR↓ | R@1↑ | R@5↑ | R@10↑ | MdR↓ |
| CE Liu et al. (2019) | 14.9 | 40.2 | 52.8 | 9 | 16.4 | 41.3 | 54.4 | 8.7 |
| MMT Gabeur et al. (2020) | 18.2 | 46.0 | 60.7 | 7 | 20.3 | 49.1 | 63.9 | 6 |
| MVAE Wu et al. (2018) | 17.8 | 46.5 | 61.0 | 7 | 20.1 | 48.7 | 64.2 | 6 |
| IPD (Ours) | **18.6** | **47.4** | **62.3** | **6.5** | **20.6** | **50.3** | **65.8** | **5.3** |

Table 5: Paramters of the implementation of task prototypes.

| Method | Params. (Prototypes) | Params. (All) | R@5 |
|---|---|---|---|
| w/o. All | $5.12 \times 10^2$ | $1.33 \times 10^8$ | 46.0 |
| w/o. LR | $6.56 \times 10^4$ | $1.33 \times 10^8$ | 46.4 |
| IPD (Full) | $2.05 \times 10^4$ | $1.33 \times 10^8$ | 47.4 |

as shown in Fig. 3. IPD with contrastive constraints could achieve competitive performances based a relatively small rank values (suppose that $k = R$). When removing the contrastive constraints that can promote the multimodal association, a large rank value is needed for the satisfactory results.

We also conduct analysis for the low-rank ensemble learning. As shown in Fig. 4, IPD has poor performances on the single modality without the mechanism, which reveals its effectiveness again.

### 5.5 EXPERIMENTS FOR SPEAKER TRAITS RECOGNITION

Table 3 shows the experimental results of different methods on speaker traits recognition dataset POM, where the top half part corresponds to existing combinations and the bottom half part corresponds to unseen combinations, we report the multi-class accuracy of multiple traits[3]. The simialr observation could be found from the table, IPD achieves competitive performances on both existing and unseen modality combinations compared with the baseline methods. Particularly, the performance of IPD increases the average multi-class accuracy from 26.6 to 28.8 (existing combs.) and from 30.5 to 33.8 (unseen combs.) compared to the best counterparts.

### 5.6 EXPERIMENTS FOR VIDEO RETRIEVAL

To evaluate the generalization for more modalities, we report the evaluation results of IPD and the competing text-video retrieval methods on MSR-VTT (Table 4). Note that MMIN, MCTN, and most of the existing methods (i.e. Ji et al. (2022); Lei et al. (2021)) can not be used for 7 modalities, thus, we do not compare IPD with them. We provide the metrics from two points of views (i.e. existing and unseen modality combinations). For both settings, IPD outperforms baseline methods in all the metrics. Benefiting from the low-rank prototype decomposition and disentangled contrastive constraints, the modality-shared knowledge and modality-specific knowledge are efficiently combined for the corresponding tasks. Further, the complexity reduction is more obvious for video retrieval, due to the larger modality number and hidden size. To provide an intuitive of complexity reduction, we report the parameter number (Table 5) of the task prototypes.

## 6 CONCLUSION

We propose a novel method called Interaction Augmented Prototype Decomposition (IPD) to solve the generalizable missing modality problem. Concretely, IPD disentangles the task prototypes into a modality-shared part and a low-rank modality-specific part. We present a principled analysis to provide the rationality of low-rank approximation. Further, to control the rank value, we facilitate the interaction of different modality conditions by employing disentangled contrastive constraints, which complements the decomposition. Extensive results on the newly-created benchmarks of multiple tasks illustrates the superiority of our proposed method.

---

[3]Due to the space limits, we provide the results of 4 traits, the complete results are shown in the appendix.

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

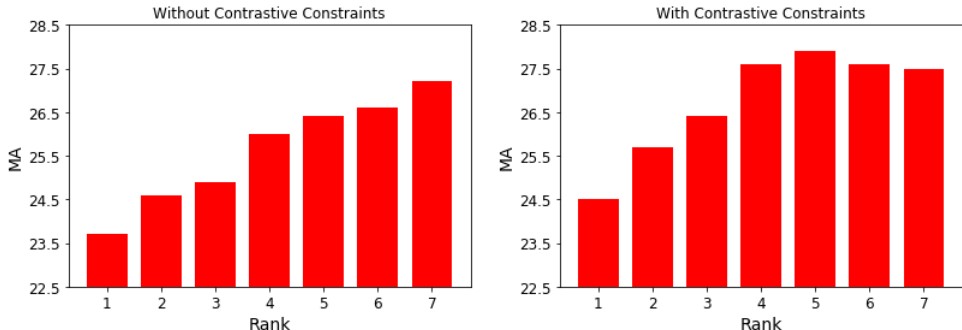

Figure 3: Ablation study of rank value, suppose that two rank values ($k$ in Eq. 5 and $R$ in Eq. 6) are equal. Left and right parts denote IPD without and with contrastive constraints, respectively.

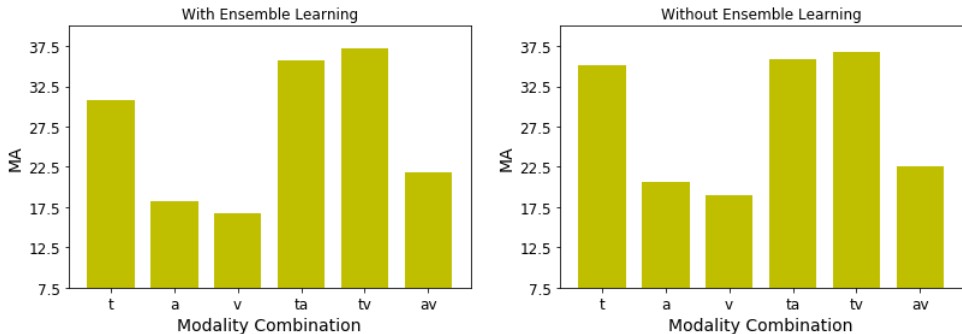

Figure 4: Ablation study of low-rank ensemble learning, where we report the multi-class accuracy of different modality conditions. Left and right parts denote IPD without and with ensemble learning.

AmirAli Bagher Zadeh, Paul Pu Liang, Soujanya Poria, Erik Cambria, and Louis-Philippe Morency. Multimodal language analysis in the wild: Cmu-mosei dataset and interpretable dynamic fusion graph. In *Proceedings of the 56th Annual Meeting of the Association for Computational Linguistics (Volume 1: Long Papers)*, pp. 2236–2246, 2018b.

Jinming Zhao, Ruichen Li, and Qin Jin. Missing modality imagination network for emotion recognition with uncertain missing modalities. In *ACL*, 2021.

Bolei Zhou, Agata Lapedriza, Aditya Khosla, Aude Oliva, and Antonio Torralba. Places: A 10 million image database for scene recognition. *IEEE transactions on pattern analysis and machine intelligence*, 2017.

Kaiyang Zhou, Chen Change Loy, and Ziwei Liu. Semi-supervised domain generalization with stochastic stylematch. *arXiv preprint arXiv:2106.00592*, 2021.

Tianfei Zhou, Wenguan Wang, Ender Konukoglu, and Luc Van Gool. Rethinking semantic segmentation: A prototype view. In *Proceedings of the IEEE/CVF Conference on Computer Vision and Pattern Recognition*, pp. 2582–2593, 2022.

## A  PROOF AND DERIVATION

### A.1  THE PROOF OF PROPOSITIONS

**Proposition 1:** When $e_c \perp \text{Span}(A_s)$, the decomposition of $E = e_c \mathbf{1}^\top + A_s \beta^\top$ has a unique solution.

**Proof:** Suppose $E = e_c \mathbf{1}^\top + A_s \beta^\top = w_c \mathbf{1}^\top + W_s \Gamma^\top$ is a rank-$(k+1)$ matrix, where $A_s \in \mathbb{R}^{m \times k}$, $\beta \in \mathbb{R}^{2^n \times k}$, $W_s \in \mathbb{R}^{m \times k}$, and $\Gamma \in \mathbb{R}^{2^n \times k}$ are all rank-$k$ matrices with $k \leq \min(m, 2^n)$. When

$e_c \perp \text{Span}(A_s)$, to prove that $w_c = e_c$ is equal to $w_c \perp \text{Span}(W_s)$. We should deduce the necessary and sufficient conditions.

**Sufficient Condition:** Suppose that $w_c \perp \text{Span}(W_s)$. Then $E^\top w_c = \langle e_c, w_c \rangle \cdot \mathbf{1} + \beta \cdot \left( A_s^\top w_c \right) = \|w_c\|^2 \cdot \mathbf{1}$. Since $E$ is a rank-$(k+1)$ matrix, we know that $\mathbf{1} \notin \text{Span}(\beta)$ and so it has to be the case that $\langle e_c, w_c \rangle = \|w_c\|^2$ and $A_s^\top w_c = 0$. Both of these together imply that $w_c$ is the projection of $e_c$ onto the space orthogonal to $A_s$ i.e., $w_c = e_c - P_{A_s} e_c$, where $P_{A_s}$ is the projection matrix onto the span of vectors $A_s$. Since $e_c \perp A_s$, we can obtain that $w_c = e_c$.

**Necessary Condition:** Let $w_c = e_c$. Since $E$ is fixed, we could obtain $A_s \beta^\top = W_s \Gamma^\top$, i.e. $\text{Span}(W_s) = \text{Span}(A_s)$. Since $e_c \perp \text{Span}(A_s)$, $w_c \perp \text{Span}(W_s)$.

**Proposition 2:** If the orthogonal regularization is not satisfied ($e_c$ is not orthogonal to $\text{Span}(A_s)$), the performance of partial modality combinations will be influenced.

**Proof:** Suppose the modality-specific prototypes of two single modalities are $e_1$ and $e_2$ ($e_1 \perp e_2$), the fusion representation of input sample (corresponding to $e_2$) is $u$. If the orthogonal regularization is not satisfied (i.e., $e_1$ is not orthogonal to $e_c$). The prediction can be expressed as $u \cdot e_c + u \cdot e_2$. In the extreme situation (i.e., $e_1 = e_c$), the result is $u \cdot e_2$. The value of $u \cdot e_c$ is influenced by the nonorthogonality, to some extent.

## A.2 THE DERIVATION OF EQ. 9

Note that we omit $M$ for convenience. To prove the calculation of $O_s$ as follows:

$$O_s = \text{sum}\left(V \odot \left(A_s \beta^\top\right)\right) = \text{sum}\left(\Big[\sum_{r=1}^{R}\prod_{i=1}^{n} v_i C_{i,r}\Big] \odot A_s\right) \tag{10}$$

for convenience, we adopt the element-wise multiplication. Concretely, $V \in \mathbb{R}^{m \times 2^n}$ and $A_s \beta^\top \in \mathbb{R}^{m \times 2^n}$ cannot be operated with matrix (vectorial) multiplication. Therefore, we indirectly calculate the result of $(V)_d \in \mathbb{R}^{2^n}$ and $(A_s \beta^\top)_d \in \mathbb{R}^{2^n}$, $d \in [m]$, we rewrite the above equations,

$$
\begin{aligned}
o_s &= \left(A_{s,d} \beta^\top\right)\Big(\bigotimes_{i=1}^{n}(v_i)_d\Big) = A_{s,d}\Big(\beta^\top \Big(\bigotimes_{i=1}^{n}(v_i)_d\Big)\Big) \\
&= \Big(\Big(\bigotimes_{i=1}^{n}(v_i)_d\Big)^\top \beta\Big) A_{s,d}^\top = \Big(\Big(\bigotimes_{i=1}^{n}(v_i)_d\Big)^\top \big[C^1, ..., C^k\big]\Big) A_{s,d}^\top
\end{aligned}
\tag{11}
$$

where $(v_i)_d \in \mathbb{R}^2$, $A_{s,d} \in \mathbb{R}^k$. We then calculate $\Big(\bigotimes_{i=1}^{n}(v_i)_d\Big)^\top C^j$ as follows:

$$\Big(\bigotimes_{i=1}^{n}(v_i)_d\Big)^\top C^j = \Big(\bigotimes_{i=1}^{n}(v_i)_d\Big)\Big(\sum_{r=1}^{R}\bigotimes_{i=1}^{n} C_{i,r}^j\Big) = \sum_{r=1}^{R}\prod_{i=1}^{n}(v_i)_d C_{i,r}^j \tag{12}$$

we put the result of Eq. 12 into Eq. 11 and obatain:

$$o_s = \Big(\sum_{r=1}^{R}\prod_{i=1}^{n}(v_i)_d C_{i,r}\Big) A_{s,d} = \text{sum}\Big(\Big[\sum_{r=1}^{R}\prod_{i=1}^{n}(v_i)_d C_{i,r}\Big] \odot A_{s,d}\Big) \tag{13}$$

According to the conclusion of Eq. 13, Eq. 10 is proved. We then prove the the calculation of $O_c$ as follows:

$$O_c = \text{sum}\left(V \odot \left(e_c \mathbf{1}^\top\right)\right) = \text{sum}\left(\Big[\prod_{i=1}^{n} v_i \cdot \mathbf{1}\Big] \odot e_c\right) \tag{14}$$

We also adopt element-wise multiplication.

$$o_c = \left(\left(\bigotimes_{i=1}^{n} (v_i)_d\right) \cdot \mathbf{1}^{2^n}\right) e_{c,d} = \text{sum}\left(\left[\prod_{i=1}^{n} (v_i)_d \cdot \mathbf{1}^2\right] \odot e_{c,d}\right) \tag{15}$$

According to the conclusion of Eq. 15, Eq. 14 is proved.

---

**Algorithm 1** The Training Process of IPD

---

1: **Given:** $n, m, k, \lambda_1, \lambda_2, \lambda_3$, train-data
2: Initialize params $e_c \in \mathbb{R}^m, A_s \in \mathbb{R}^{m \times k}$
3: Initialize $C_{i,r}^j \in \mathbb{R}^2 : i \in [n], r \in [R], j \in [k]$ in Eq. 6
4: $\mathcal{L}_o \leftarrow \|I_{k+1} - [e_c, A_s]^\top [e_c, A_s]\|^2$             ▷ Orthonormality constraint
5: **for** $(x, y, l) \in$ train-data **do**
6:      $E \leftarrow e_c \mathbf{1}^\top + A_s \beta^\top$             ▷ $\beta_l$ and $\beta$ are calculated based on Eqs. 5 and 6
7:      $e_l \leftarrow e_c + A_s \beta_l^\top$
8:      loss $+= \mathcal{L}_t(y^*(x), y; e_l) + \lambda_1 \mathcal{L}_e(O_e(x), y; E) + \lambda_2 \mathcal{L}_o + \lambda_3 \mathcal{L}_c$
9: **end for**
10: Optimize loss wrt $e_c, A_s, C_{i,r}^j$
11: **Return** $e_c, A_s, C_{i,r}^j$

---

---

**Algorithm 2** The Inference Process of IPD

---

1: **Given:** $n, m, k$, test-data
2: Trained params $e_c \in \mathbb{R}^m, A_s \in \mathbb{R}^{m \times k}, \beta \in \mathbb{R}^{2^n \times k}$     ▷ $\beta$ is calculated based on Eqs. 5 and 6
3: **for** $(x, l) \in$ test-data **do**
4:      $e_l \leftarrow e_c + A_s \beta_l$
5:      $y^*(x) = \text{Net}(x; e_l)$
6: **end for**

---

# B MORE EXPERIMENTAL DETAILS

## B.1 THE DETAILED PROCESS OF IPD

Due to the space limitation, we put the algorithmic process (both training and testing) in this section. The mathematic symbols are same as the main paper. In the training process, $O_e(x)$ denotes the output $O_e$ of input $x$, $\mathcal{L}_e(O_e(x), y; E)$ denotes the calculated loss based on modality factors $E$, and $\mathcal{L}_t(y^*(x), y; e_l)$ denotes the calculated loss based on modality factor $e_l$.

In the testing process, we first obtain the modality factor $e_l$ according to the modality combination index, then, we send the input $x$ to the network and obtain $y^*(x) = \text{Net}(x; e_l)$.

## B.2 THE TRAINING OBJECTIVE OF VIDEO RETRIEVAL

As shown in MMT Gabeur et al. (2020), the training objective of video retrieval is triplet contrastive loss, which compares the matching scores between video representation and text representation. To implement our method, we employ a new contrastive cross-entropy loss. Concretely, we treat the query text as one of the modalities of the video and text modality always exists. If the query text and the video are matched, the classification output is 1. In a batch, we treat the matched text and video as positive sample and change one side to construct negative samples. Then, the cross-entropy loss can be used to train the network.

## B.3 THE FEATURE EXTRACTION FOR VIDEO RETRIEVAL

Motion embeddings are extracted from S3D Xie et al. (2018) trained on the kinetics dataset. Scene embeddings are extracted with DenseNet-161 Huang et al. (2017) trained on the Places365 dataset Zhou et al. (2017). OCR embeddings are extracted in three stages. First, the pixel link text detection

model is used to detect the overlaid text. Then, the detected boxes are passed through the text recognition model. Finally, each character sequence is encoded using a word2vec embedding. Audio embeddings are obtained with a VGGish model, trained on the YouTube-8m dataset. Speech features are extracted using the Google Cloud speech API, to extract word tokens from the audio stream, which are then encoded via pre-trained word2vec embeddings Mikolov et al. (2013). Face features are extracted by ResNet-50 He et al. (2016) trained for face classification on the VGGFace2 dataset. Appearance features are extracted from the final global average pooling layer of SENet-154 Hu et al. (2018) trained on ImageNet.

### B.4 Experimental Details for Multimodal Sentiment Analysis and Speaker Traits Recognition

We follow LMF to implement IPD. All the components of IPD are same as those of LMF, except for the final classification layer. Specifically, the hidden size of the task prototype is 16, the rank $k$ and $R$ are set to 5. $\Delta$ is set to 0.1. The tuning of other parameters is similar to LMF. After grid searching, the batch size is set to 16, the learning rate is 0.001. The hidden size of common feature space is 16. $\lambda_1, \lambda_2, \lambda_3$ are set to 0.1. We develop all the experiments with 5 RTX-3080Ti GPUs (10GB).

### B.5 Experimental Details for Video Retrieval

We follow MMT to implement IPD. All the components of IPD are same as those of MMT. We add the new contrastive cross-entropy loss introduced above to the module and employ it as an auxiliary loss. Specifically, the hidden size of the task prototype is 512, the rank $k$ and $R$ are set to 32. $\Delta$ is set to 0.1. The tuning of other parameters is similar to MMT. Concretely, the batch size is 32, the initial learning rate is set to $5e-5$, which decays by a multiplicative factor 0.95 every 1k optimization steps. We train for 50k steps. The hidden size is 512 for all the Transformer structures, the number of heads is 8, and there are 6 stacked attention blocks. $\lambda_1, \lambda_2, \lambda_3$ are set to 0.1. We develop all the experiments with 5 RTX-3080Ti GPUs (10GB).

### B.6 Experimental Details of Baseline Methods

We simply reproduce the baseline methods by replace the missing-modality features by all-zero vectors and follow these methods for the subsequent processes.

## C More Experimental Results

### C.1 Additional Results of POM

The results of other all 16 traits are shown in Table 6 and Table 7. We could obtain similar conclusions.

### C.2 Additional Results of The Case That All The Modalities Are Available When Training

We also conduct experiments when all the modalities are available during training. In this way, we adopt the same train/val/test splits as LMF. The validation and test sets are equally divided into 7 pieces of modality combinations. We report the corresponding results of different methods. As shown in the figure, the metric gap between IPD and baseline methods narrows, since MVAE, MCTN, MMIN are original designed for this case.

### C.3 Additional Results of CMU-MOSEI

The results of CMU-MOSEI Zadeh et al. (2018b) are shown in Table 9. CMU-MOSEI is also proposed for the evaluation of multimodal sentiment analysis. It is similar to CMU-MOSI and has 23454 movie review videos. We could obtain similar conclusions based on Table 9.

Table 6: The performances on POM for existing modality combinations. MA(5,7) denotes multi-class accuracy for (5,7) classes.

| Method | Con | Pas | Voi | Dom | Cre | Viv | Exp | Ent |
|---|---|---|---|---|---|---|---|---|
| | MA7 | MA7 | MA7 | MA7 | MA7 | MA7 | MA7 | MA7 |
| TFN Zadeh et al. (2017) | 21.7 | 26.1 | 20.6 | 27.1 | 23.2 | 23.2 | 19.7 | 24.5 |
| MFN Zadeh et al. (2018a) | 20.6 | 25.5 | 23.2 | 28.5 | 19.6 | 23.6 | 20.7 | 26.1 |
| LMF Liu et al. (2018) | 23.6 | 26.1 | 24.5 | 29.5 | 19.6 | 23.2 | 20.7 | 28.1 |
| MulT Tsai et al. (2019) | 23.6 | 27.6 | 28.6 | 29.5 | 23.2 | 23.6 | 22.7 | 28.1 |
| MVAE Wu et al. (2018) | 21.7 | 20.2 | 28.6 | 28.5 | 23.2 | 23.6 | 25.1 | 26.1 |
| MCTN Pham et al. (2019) | 23.6 | 27.6 | 28.6 | 28.5 | 23.2 | 23.2 | 20.7 | 24.5 |
| MMIN Zhao et al. (2021) | 21.7 | **27.6** | 30.5 | 30.0 | 25.6 | 27.6 | 25.1 | 28.1 |
| IPD (Ours) | **25.6** | 27.1 | **34.5** | **31.5** | **27.1** | **30.5** | **27.1** | **30.5** |

| Method | Res | Tru | Rel | Out | Tho | Ner | Per | Hum |
|---|---|---|---|---|---|---|---|---|
| | MA5 | MA5 | MA5 | MA5 | MA5 | MA5 | MA7 | MA5 |
| TFN Zadeh et al. (2017) | 22.7 | 37.4 | 36.9 | 37.4 | 36.5 | 35.5 | 36.0 | 37.4 |
| MFN Zadeh et al. (2018a) | 25.6 | 37.9 | 38.4 | 38.9 | 35.5 | 35.5 | 37.4 | 38.4 |
| LMF Liu et al. (2018) | 25.1 | 38.9 | 38.9 | 37.4 | 36.9 | 34.0 | 36.9 | 37.4 |
| MulT Tsai et al. (2019) | 27.6 | 39.9 | 38.9 | 37.4 | 37.9 | 35.5 | 36.9 | 37.4 |
| MVAE Wu et al. (2018) | 26.6 | 37.9 | 40.4 | 38.4 | 37.9 | 37.4 | 38.9 | 39.9 |
| MCTN Pham et al. (2019) | 27.6 | 39.9 | 40.9 | 40.9 | 38.4 | 37.9 | 38.9 | 40.4 |
| MMIN Zhao et al. (2021) | 29.6 | 39.9 | 40.4 | 41.9 | 38.4 | 39.4 | 39.9 | 41.9 |
| IPD (Ours) | **29.6** | **39.9** | **41.9** | **43.8** | **39.4** | **39.4** | **40.9** | **42.4** |

Table 7: The performances on POM for unseen combination. MA(5,7) denotes multi-class accuracy for (5,7) classes.

| Method | Con | Pas | Voi | Dom | Cre | Viv | Exp | Ent |
|---|---|---|---|---|---|---|---|---|
| | MA7 | MA7 | MA7 | MA7 | MA7 | MA7 | MA7 | MA7 |
| TFN Zadeh et al. (2017) | 27.6 | 32.0 | 27.1 | 32.5 | 28.1 | 27.6 | 25.6 | 29.6 |
| MFN Zadeh et al. (2018a) | 26.1 | 31.5 | 29.6 | 34.5 | 25.6 | 29.6 | 26.6 | 31.0 |
| LMF Liu et al. (2018) | 28.6 | 32.0 | 30.5 | 35.5 | 25.6 | 27.6 | 26.6 | 33.5 |
| MulT Tsai et al. (2019) | 27.6 | 31.5 | 34.5 | 34.5 | 28.1 | 29.6 | 27.6 | 33.5 |
| MVAE Wu et al. (2018) | 27.6 | 27.1 | 34.5 | 34.0 | 29.6 | 29.6 | 27.6 | 31.0 |
| MCTN Pham et al. (2019) | 28.6 | 33.5 | 34.5 | 34.0 | 29.6 | 27.6 | 25.6 | 31.5 |
| MMIN Zhao et al. (2021) | 27.6 | 33.5 | 34.5 | 35.5 | 30.5 | 30.5 | 27.6 | **33.5** |
| IPD (Ours) | **30.5** | **33.5** | **36.9** | **37.4** | **34.5** | **36.9** | **31.0** | 33.0 |

| Method | Res | Tru | Rel | Out | Tho | Ner | Per | Hum |
|---|---|---|---|---|---|---|---|---|
| | MA5 | MA5 | MA5 | MA5 | MA5 | MA5 | MA7 | MA5 |
| TFN Zadeh et al. (2017) | 27.6 | 41.9 | 42.9 | 44.8 | 40.9 | 38.4 | 39.9 | 42.4 |
| MFN Zadeh et al. (2018a) | 29.6 | 39.9 | 42.9 | 45.8 | 39.9 | 38.9 | 41.9 | 43.8 |
| LMF Liu et al. (2018) | 28.6 | 41.4 | 43.8 | 44.8 | 39.4 | 38.4 | 40.9 | 43.8 |
| MulT Tsai et al. (2019) | 31.0 | 43.8 | 43.8 | 44.8 | 40.9 | 38.9 | 41.4 | 42.4 |
| MVAE Wu et al. (2018) | 30.5 | 43.3 | 44.3 | 45.3 | 40.4 | 40.9 | 41.9 | 44.3 |
| MCTN Pham et al. (2019) | 30.5 | 43.8 | 45.3 | 46.8 | 40.9 | 41.4 | 41.4 | 44.8 |
| MMIN Zhao et al. (2021) | 31.0 | **44.8** | 45.3 | 47.8 | 40.4 | 41.4 | 40.9 | 44.3 |
| IPD (Ours) | **32.5** | 43.8 | **46.8** | **48.3** | **41.9** | **42.4** | **42.9** | **44.8** |

## C.4 ADDITIONAL RESULTS OF RGB-D AND XRMB

We also conduct experiments on XRMB and RGB-D. As for XRMB, it has two modalities (273D acoustic inputs and 112D articulatory inputs) following Wang et al. (2015), thus, the power of IPD is not obvious with achieving a similar results to baseline methods. As for RGB-D, due to the imbalanced importance of different modalities (3D point cloud, RGB color and height, following

Table 8: The results on CMU-MOSI when all the modalities are available when training.

| Method | Metrics | | | | |
|---|---|---|---|---|---|
| | BA | F1 | MAE | Corr | MA |
| MVAE Wu et al. (2018) | 74.3 | 74.5 | 1.065 | 0.621 | 27.8 |
| MCTN Pham et al. (2019) | 73.9 | 74.0 | 1.019 | 0.601 | 28.9 |
| MMIN Zhao et al. (2021) | 74.8 | 74.7 | 0.987 | **0.625** | 29.6 |
| IPD (Ours) | **75.4** | **75.7** | **0.985** | 0.621 | **30.2** |

Table 9: The results on CMU-MOSEI. We report BA (binary accuracy), F1, Corr (Correlation Coefficient), MA (Multi-class accuracy, higher is better), MAE (Mean-absolute Error, lower is better).

| Method | Existing Combs. | | | | | Unseen Combs. | | | | |
|---|---|---|---|---|---|---|---|---|---|---|
| | BA | F1 | MAE | Corr | MA | BA | F1 | MAE | Corr | MA |
| MVAE Wu et al. (2018) | 67.2 | 67.5 | 1.050 | 0.409 | 36.7 | 76.1 | 75.7 | 0.725 | 0.498 | 43.5 |
| MCTN Pham et al. (2019) | 66.9 | 67.3 | 1.007 | 0.440 | 36.4 | 74.9 | 75.0 | 0.735 | 0.497 | 43.5 |
| MMIN Zhao et al. (2021) | 67.5 | 67.8 | 0.975 | 0.429 | 37.3 | 75.7 | 75.0 | 0.782 | 0.508 | 42.9 |
| IPD (Ours) | **69.8** | **70.2** | **0.909** | **0.470** | **39.6** | **77.8** | **78.4** | **0.712** | **0.515** | **45.0** |

Liu et al. (2021)), we keep the 3D point cloud as a fixed available modality. The missing modality setting is adopted to RGB color and height. Since it is hard to model multimodal interaction with two modalities, IPD also performs similar to the baseline methods.

Table 10: The results on XRMB. We report PER (phone error rate).

| Method | Existing Combs. | Unseen Combs. |
|---|---|---|
| | PER | PER |
| MVAE Wu et al. (2018) | 0.274 | 0.255 |
| MCTN Pham et al. (2019) | 0.272 | 0.248 |
| MMIN Zhao et al. (2021) | 0.283 | 0.252 |
| IPD (Ours) | 0.267 | 0.245 |

Table 11: The results on RGB-D. We report mAP@0.25 for 3D object detection.

| Method | Existing Combs. | Unseen Combs. |
|---|---|---|
| | mAP@0.25 | mAP@0.25 |
| MVAE Wu et al. (2018) | 43.6 | 49.2 |
| MCTN Pham et al. (2019) | 44.7 | 49.5 |
| MMIN Zhao et al. (2021) | 45.8 | 49.8 |
| IPD (Ours) | 46.7 | 50.4 |

