# OpenReview forum: "Rethinking Missing Modality Learning: From a Decoding View"
_ICLR.cc/2023/Conference — Submitted to ICLR 2023_

### Official Review · Reviewer_QUyJ · 2022-10-24

**Confidence:** 3
**Correctness:** 3
**Technical Novelty And Significance:** 3
**Empirical Novelty And Significance:** 3
**Recommendation:** 5

**Clarity, Quality, Novelty And Reproducibility:**

Some figures are unclear. The authors should do more experiments on different datasets, including XRMB and RGB-D. Besides, the authors can public their code for reproducibility.

**Strength And Weaknesses:**

The idea is interesting. However, the motivation is not clearly given. Besides, the writing can be improved.

**Summary Of The Paper:**

This paper proposes an Interaction Augmented Prototype Decomposition (IPD) model for missing modality learning.

**Summary Of The Review:**

This paper proposes an Interaction Augmented Prototype Decomposition (IPD) model for missing modality learning. Some key points are easy to follow. However, the clarity needs to be improved.

---

> ### Author Response · Authors · 2022-11-16
> **The response for Reviewer QUyJ**
>
> **1. About the motivation**
>
> Thanks for the advice. We chronically illustrate our motivation (i.e., why the proposed module is effective for the performance) in each section of the corresponding module. For the convenience of reading, we have red-color-coded these motivations.
>
> **2. About the clear Figures**
>
> Thanks for the advice. We have polished the main figures in the paper, by enlarging the words, enhancing the connection of different modules and corresponding equations, and adding more detailed illustration. The reviewer can refer to the polished figures in the paper.
>
> **3. About the more experiments**
>
> Thanks for the advice. We conduct experiments on other three datasets in the paper. Following the advice, we supplement the experiments on XRMB, RGB-D, and CMU-MOSEI in the appendix (Table 9, 10, 11). The reviewer can refer to them. Since the number of operated modalities of XRMB, RGB-D is two (little interaction of different modality combinations), the advantage of IPD is not obvious.
>
> **4. About the codes**
>
> Thanks for the advice. Based on the openreview platform, we promise that the codes must be released once the paper is accepted.

---

> ### Author Response · Authors · 2022-12-10
> **We would like to hear back from Reviewer QUyJ**
>
> Dear Reviewer QUyJ,
>
> We would like to follow up to see if our response addresses your concerns or if you have any further questions. We would really appreciate the opportunity to discuss this further if our response has not already addressed your concerns. Thank you again!

---

### Official Review · Reviewer_YKCH · 2022-10-24

**Confidence:** 4
**Correctness:** 3
**Technical Novelty And Significance:** 2
**Empirical Novelty And Significance:** 2
**Recommendation:** 3

**Clarity, Quality, Novelty And Reproducibility:**

*Clarity:* In addition to the difficulties understanding the technical details described above, I also found the overall exposition quite difficult to follow. There is no dataflow diagram, model description, or algorithm explicitly describing the proposed method. The authors make frequent reference to “LMF” without providing any hint to the reader how that algorithm works. The writing style varies considerably from paragraph to paragraph.

*Quality:* I think the submission is of low quality. In addition to the lack of clarity, it seems that the authors have assembled several existing ideas (low rank decomposition, orthogonal decompositions, etc.) and shown that they work well together. But they give little explanation as to why they work well. They offer some vague generalities (e.g. “Intuitively, the common component contains more high-level semantic information (i.e., the reasoning of performance), and the modality-specific part contains more low-level detailed information (i.e., the loudness of voice, the movement range” (p4)), but no rigorous empirical or theoretical justification to explain the choices that were made (this includes Proposition 2, which offers little in the way of understanding).

*Novelty:* No single part of the pipeline appears to be novel, but the combination may be.

*Reproducibility:* Several key details are missing that would make it difficult to reproduce the results in this paper. Namely, the precise breakdown of the datasets into training/validation/testing sets, the optimization algorithm used and it’s hyperparameters, and how the data was packaged into (x,y) pairs.


**Strength And Weaknesses:**

*Strengths:* Dealing with missing modalities during both training and testing is an important practical problem that deserves study. The proposed method outperforms other recent methods on a benchmark introduced by the authors.

*Weaknesses:* I found this paper very difficult to follow. After several close readings, I am still not confident that I understand the technical details of the proposed method. Specifically,

(1) Are the pairs (x, y) the training data? If so, how is the input x formed?

(2) Is Equation (1) the abstract model of data generation? If so, why is that a reasonable model? If not, then where is Equation (1) coming from?

(3) How does the “maximum suppression” scheme described in Section 3.5 stop Equation (8) from blowing up exponentially?

(4) Is the ensemble learning scheme of Section 3.6 used during training only or also during testing?


**Summary Of The Paper:**

The paper under review considers the problem of multimodal learning in the case where one or more of the modalities might be unavailable during either training or testing. The proposed solution works by decomposing representation space into a collection of orthogonal vectors, a “common” component to capture high-level semantic information, and modality-specific components to capture low-level information associated with each modality.

**Summary Of The Review:**

I recommend rejection for this paper. The proposed method may work well, but I think the exposition is too difficult to follow for readers to learn much from this work. I suggest that the authors redraft this work with a focus on clarity of method, clarity of data, and clear understanding of why the proposed method works well.

---

> ### Author Response · Authors · 2022-11-16
> **The response for Reviewer YKCH (1)**
>
> **1. The Weakness**
>
> **1.1.	The pairs of training pairs (x, y)**
>
> Thanks for the advice. The pairs (x, y) are the training data of our work. Since the input of our task is video with multimodal information, we utilize LMF and MMT to transform the video into single-vector representation x. The concrete process can be expressed as x = LMF/MMT (video). LMF and MMT can be treated as plug-and-play functions.
>
> **1.2.	Is Equation (1) the abstract model of data generation?**
>
> Thanks for the advice. We polish the equation with a prediction way as $y = x^{\top} e_l$. Thus, Equation 1 is not about data generation, it illustrates the vanilla prediction process. Specifically, $x$ is the final multimodal representation, $e_l$ is the classification weights/task prototypes, $y$ is the prediction result.
>
> **1.3.	About the maximum suppression scheme**
>
> Thanks for the advice. As we illustrate in the paper, suppose that we have three modalities and the possible combinations include {t, a, v, ta, tv, av, tav}. For example, when we choose the positive and negative pairs for the anchor {t}, there are multiple positive and negative pairs, i.e., {t, ta, a} (ta is positive and a is negative), {t, ta, v}, {t, ta, av}, {t, tv, a}, {t, tv, v}, {t, tv, av}, {t, tav, a}, {t, tav, v}, {t, tav, av}. The total number of these pairs are $(2^{n-1}-1)*(2^{n-1}-1)$, where $n$ denotes the modality number. Therefore, it scales exponential to the $n$. When we utilize the maximum suppression scheme, the positive and negative pair of anchor {t} becomes {t, tav, av}, since only the combination {tav} with most modalities is treated as positive object, {av}, as the complementary combination {tav}-{t}, is treated as negative object.
>
> **1.4.	Is the ensemble learning scheme of Section 3.6 used during training only or also during testing**
>
> Thanks for the advice. We illustrate this problem in Section 4, the ensemble learning is only utilized for training. Further, we also provide the detailed algorithm flow chart (Algorithm 1 and 2) in the appendix (Section B.1). The reviewer can refer to it.
>
> **2. The Clarity**
>
> **2.1.	About the baseline method LMF**
>
> Thanks for the advice and sorry for neglecting the illustration of baseline method. The details of LMF are not very important to some extent, we can simply treat it as a plug-and-play function like x = LMF(video). The main method of our work is decomposing $e_l$.
>
> **2.2.	About the clear illustration of process**
>
> Thanks for the advice. We recommend that the reviewer can read this work by combining the Algorithm flow chart in the appendix (Algorithm 1 and Algorithm 2, Section B.1). In the main paper, we illustrate the process of overall framework (i.e., dataflow diagram, model description) of Fig. 2, while the concrete implementation process is shown in the appendix (Algorithm 1 and Algorithm 2). We also have polished the Fig. 2 with more explanations, like the correspondence between equations/variables and modules.
>
> **3. The Quality**
>
> **3.1.	About little explanation of why work**
>
> Thanks for the advice. We chronically illustrate our motivation (i.e., why the proposed module is effective for the performance) in each section of the corresponding module. Most of these motivations are from conventional empirical. In addition, we also conduct many ablation experiments (Sec 5.4 and 5.5) to evaluate these motivations. To some extent, we argue the explanation which contains both empirical and experimental is tough. Besides, for the convenience of reading, we have red-color-coded these motivations.
>
> **3.2.	About the claim “the common component contains more high-level semantic information”**
>
> Thanks for the advice. We utilize this claim intuitively (i.e., from conventional empirical). For example, in the task of multimodal sentiment analysis, the input is a video of talking face. Thus, we utilize three modality, visual pose, speaking audio, concrete text. The only common info of these three modality is the speaking content, which can be reflected by all of them. However, much info is owned by specific modality, like the pose amplitude, audio pitch.
>
> **4. The Novelty**
>
> Thanks for the advice. Although the tensor decomposition (like Transformer) is widely used in real-world applications, the task-specific improved version can also be novel to some extent. In this paper, we model the interaction between different modality combination by tensor decomposition and recovery, it is the first time to implement this mechanism. Besides, the modality interaction is further enhanced by the contrastive constraints. In addition, the low-rank ensemble learning comes up with the intermediate products of tensor decomposition.
>
> Therefore, the contributions of this paper are all original and task-specific.
>
> .

---

> > ### Author Response · Authors · 2022-11-16
> > **The response for Reviewer YKCH (2)**
> >
> > **5. The reproduction**
> >
> > Thanks for the advice. The Adam optimizer ($\beta_1=0.9, \beta_2=0.998$) is used for all the tasks. The learning rate is $1 \times 10^{-3}$ for sentiment analysis and trait recognition, and $5 \times 10^{-5}$ for video retrieval. **For more details, the reviewer can refer to Section B.4 and B.5.** The video data are first sent to the feature extractor (which is introduced in Section 5.2), the multimodal features are sent to LMF for downstream tasks, which is regarded as x = LMF(video). As for the dataset split, we provide the detailed process in Section 5.3, the samples are grouped into groups based on the random generated number.

---

> ### Author Response · Authors · 2022-12-10
> **We would like to hear back from Reviewer YKCH**
>
> Dear Reviewer YKCH,
>
> We would like to follow up to see if our response addresses your concerns or if you have any further questions. We would really appreciate the opportunity to discuss this further if our response has not already addressed your concerns. Thank you again!

---

### Official Review · Reviewer_pCS1 · 2022-10-25

**Confidence:** 4
**Clarity, Quality, Novelty And Reproducibility:** Paper is largely clear. Some concerns…
**Correctness:** 3
**Technical Novelty And Significance:** 2
**Empirical Novelty And Significance:** 2
**Recommendation:** 5

**Strength And Weaknesses:**

Strengths:
1. The paper is well motivated and described. The ideas are clear and experiments are on several large multimodal datasets.
2. The paper is largely clear with clear figures and exposition.

Weaknesses:
1. There are insufficient comparisons to work in missing modalities, with 'Jinming Zhao, Ruichen Li, and Qin Jin. Missing modality imagination network for emotion recognition with uncertain missing modalities. In ACL, 2021.' being the only one. The authors cite 'Paul Pu Liang, Zhun Liu, Yao-Hung Hubert Tsai, Qibin Zhao, Ruslan Salakhutdinov, and LouisPhilippe Morency. Learning representations from imperfect time series data via tensor rank regularization. In ACL 2019.' but do not compare to it (they also use low rank decomposition) of multimodal representations, and there are also some simple baselines such as including modality dropout during training that should also be compared to.
2. There should be more comparisons in both performance and complexity - the proposed method does better but certainly uses more computation so the tradeoff should be analyzed.

**Summary Of The Paper:**

This paper studies the problem of robustness in the face of noisy or missing modalities and proposes a method called Interaction Augmented Prototype Decomposition (IPD) for the setting where the number of modalities is arbitrary and there are various incomplete modality conditions happening in both training and inference phases. Their approach jointly learns the common and modality-specific task factors via low-rank decomposition, which seems to promote unseen generalization in empirical experiments. They show strong results on 3 major multimodal datasets.

**Summary Of The Review:**

Some additional comparisons and tradeoff analysis would be good.

---

> ### Author Response · Authors · 2022-11-16
> **The response for Reviewer pCS1**
>
> **1. About the weakness**
>
> **1.1.	About the compared baselines**
>
> Thanks for the advice. T2FN of ACL 19 mainly processes the problem of missing feature channels, thus, we do not compare with it directly. However, the implementation is easy to some extent. We add the results of T2FN and random drop in the paper. The reviewer can refer to polished Table 1 and Table 3. We blue-color-coded T2FN and random drop.
>
> **1.2.	About the tradeoff between performance and complexity**
>
> Thanks for the advice. The reviewer can refer to polished Table 5. We take the video retrieval for example. Although directly sharing all the networks has the minimum computation (**w/o. All**), it has unacceptable performance. Gradually, we separate the classifier for each modality combination (**w/o. LR**), the performance is improved. However, the computational computation also increases a lot. Therefore, we adopt IPD via tensor decomposition to solve this problem. As a result, the performance is improved a lot, while the computation is acceptable. In summary, **the computational complexity can be neglected compared with the full model.**

---

> ### Author Response · Authors · 2022-12-10
> **We would like to hear back from Reviewer pCS1**
>
> Dear Reviewer pCS1,
>
> We would like to follow up to see if our response addresses your concerns or if you have any further questions. We would really appreciate the opportunity to discuss this further if our response has not already addressed your concerns. Thank you again!

---

### Official Review · Reviewer_SiTt · 2022-10-25

**Confidence:** 3
**Correctness:** 3
**Technical Novelty And Significance:** 3
**Empirical Novelty And Significance:** 2
**Recommendation:** 6

**Clarity, Quality, Novelty And Reproducibility:**

The paper needs to be improved for clarity (unsupported claims, not enough details are given, figure 2 needs to be improved (please see weaknesses).
The method seems to be novel, although each submethod seems to be already proposed in other multimodal papers. For example low-rank decomposition and CP decomposition was used in the original LMF paper as well (https://arxiv.org/pdf/2204.13707.pdf).
We believe that the results are reproducible.


**Strength And Weaknesses:**

> Strengths
1.	The authors try to develop a method to handle more realistic scenarios where some data modalities are missing.
2.	The authors try to make good use of all available problem information with the regularized losses (disentangled contrastive constraints, ensemble learning).
3.	They reduce the exponential complexity of the problem, which is dependent on the possible modality combinations (parameter l) through low-rank representations and CP decomposition.

> Weaknesses
1.	For the unseen modality combination: We believe that the current experiments do not test the model's performance on unseen modalities. It would make sense if a completely new modality were present or some modality was abnormal or out-of-distribution during testing. The fact that the trained models were exposed to all possible multimodal combinations except the case where all modalities are present seems not to test the unseen case but rather how much the model's performance can be improved when all modalities are available.
2.	Based on the previous argument, we would like to ask the authors to compute the performance of their method (and the rest) when all modalities are present to see the impact on the model when we have missing modalities. This will help us assess its performance compared to the other methods. Page 8 "...methods (i.e. MCTN, MMIN) depends on the existence of all the modalities to obtain the supervised information in the training stage; therefore, when the samples for training are imcomplete, there will be a big drop in performance. " these experiments could potentially help support this argument. Since the performance of MMIN is very close to IPD, it even, in some cases, outperforms it.
3.	Enrichment of related work, most of the referenced publications are from 2017,2018, with only one from 2019 (the most recent one). Also, give more details about previous papers, for example, " Liang et al. (2019) employs low-rank fusion for each time step of multi-view sequential input. " and how the proposed method compares to Liang et al. (2019).
4.	Some captions of figures are too short. Specifically, Figure 2. Not all diagram variables are explained. Figure 3: similarly, what is k, and what is R?
5.	Figures 3: We believe the figures would be more readable if there were a title above the bar graphs "With, without contrastive constraints."
6.	Figure 2: Please consider adding more details, like the equation connecting the two decoding parts. How are vectors connected to the architecture, and what are the gray cells? In general, the figure and caption need more work. It is tough to understand what the model is doing and how the different pieces connect to the figure. Even if we read the paper and then look at the figure, it isn't easy to map things from text to pictorial.
7.	Throughout the document, there exist claims/sentences that remain unsupported or not fully explained; we would like the authors to go through these sentences and explain what they hold.
a.	Page 4 " Furthermore, the decomposition of A\beta^T does not consider the fine-grained interaction among different modality combinations. " why is this the case? How does CP decomposition solve this problem? Does it?
b.	Pages 6: "We report the metrics of BA (binary accuracy), F1, Corr (Correlation Coefficient), MA (Multi-class accuracy, higher is better), and MAE (Mean-Absolute Error, lower is better). " "We report the common metrics of R@K and MdR. " Can the authors include the definition of these metrics or provide references?
c.	Page 7: "To be realistic, we randomly generate the ratio of 7 pieces (by giving each piece a number from 0 to 1 and employing normalization) ": not clear what normalization is applied and why.
d.	Page 8: " Even MulT utilizes Transformer Vaswani et al. (2017). " what's the conclusion for this sentence? What are the authors trying to say to the reader?
e.	Page 9: "Note that MMIN, MCTN, and most of the existing methods (i.e. Ji et al. (2022);Lei et al. (2021)) can not be used for 7 modalities, thus, we do not compare IPD with them." Please give more details, why not applicable.


**Summary Of The Paper:**

In this paper, the authors are interested in the multimodal learning problem. They want to focus on cases where incomplete data (not data from all modalities) are present during training. To do that, they assume that the fused vector x (output of LMF) can be given by the task classification (or regression) labels times e_l, which is the complete task prototype for all possible modality combinations L. Since (x,y) are given, the paper's main focus is how to learn the vector e_l efficiently. Finally, they test their trained models on unseen modality combinations.

**Summary Of The Review:**

We believe that it is important that the authors tried to develop a method for incomplete modality information during training. Because it is closer to real scenarios. The proposed method managed to improve the classification/regression performance when compared to other methods. But we believe that the paper can be improved and make more clear its advantages by addressing our comments in the previous sections.

------------------------------------
Post rebuttal summary:

We thank the authors for answering all of our questions.
1) We believe the authors' responses for 2.1, 2.2, and 2.3 should be part of the paper.
2) For 2.3 and the L1 normalization, the question referred to normalized data. It needs to be clarified from the text on what the L1 is applied.
3) Figure 2 is more clear now! But it can be further improved; currently, it mainly lists the definitions of each parameter. The authors could reformulate it, so it has a better flow.

After reading the other reviews and authors' responses, we kept our recommended score to 6.

---

> ### Author Response · Authors · 2022-11-17
> **The response for Reviewer SiTt (1)**
>
> **1. About the weakness**
>
> **1.1.	About the current experiments do not test the model's performance on unseen modalities**
>
> Thanks for the advice. Directly adopting the knowledge of some modalities to an unseen modality via a zero-shot manner is our final goal of multimodal learning, which can be treated as our future work.
>
> **1.2.	About the performance when all the modalities are present**
>
> Thanks for the advice. We take the multimodal sentiment analysis as example. The results are shown as Table 8 (Appendix Section C.2). In summary, MCTN, MMIN have close performance to IPD, since MCTN, MMIN are designed for the condition where all the modalities are present during training.
>
> **1.3.	About related work**
>
> Thanks for the advice. We have polished the related work and blue-color-coded them. The reviewer can refer to it.
>
> **1.4.	Some captions of figures are too short**
>
> Thanks for the advice. We have polished the figures and added more illustration. Please refer to Figs. 2 and 3.
>
> **1.5.       We believe the figures would be more readable if there were a title above the bar graphs With, without contrastive constraints**
>
> Thanks for the advice. We have polished Fig. 3.
>
> **1.6.       Figure 2: Please consider adding more details**
>
> Thanks for the advice. We have polished Fig. 2 for more convenient illustration, by enlarging the words, enhancing the connection of different modules,  connecting the equations, vectors to the modules, and adding more detailed illustration.
>
> **2. About more explanations**
>
> **2.1.	The decomposition of $A_s\beta^T$ does not consider the fine-grained interaction among different modality combinations**
>
> Since $\beta \in R^{2^n \times k}$, each modality combination (total $2^n$) corresponds to a $\beta_l \in R^{k}$. Therefore, even two modality combinations like {ta} and {tav} have a strong degree of overlap, $\beta_{ta}$ and $\beta_{tav}$ are trained separately without fine-grained interaction. However, the tensor decomposition let the $\beta_{ta}$ calculated with $C_{t,1}, C_{a,1}, C_{v,0},$ $\beta_{tav}$ calculated with $C_{t,1}, C_{a,1}, C_{v,1}$.  $\beta_{ta}$ and $\beta_{tav}$ have obvious overlap with $C_{t,1}, C_{a,1}$. In summary, with tensor decomposition, the fine-grained interaction of modality combinations is mined via the overlap of composition.
>
> **2.2.	About the metrics**
>
> Thanks for the advice. We employ all the metrics following the previous baselines for fairness.
>
> On multimodal sentiment analysis, we follow LMF to use BA (binary accuracy), F1, Corr (Correlation Coefficient), MA (Multi-class accuracy), and MAE (Mean-Absolute Error).
>
> **BA** denotes the binary accuracy, since sentiment analysis is a regression task and the score ranges from -3 to 3, we treat the scores [-3, 0] and [0, 3] as two classes. BA is computed by the number of accurate-prediction samples dividing the total number.
>
> **F1** denotes the F1 score, which is used specifically for two-class classification. It is computed with two values, precision and recall, $F1 = \frac{2 \times precision \times recall}{precision + recall}$.
>
> **Corr** denotes Correlation Coefficient, which computes the function between GT (y) and prediction result (y’). The function is defined as $\frac{Cov(y, y’)}{(\sqrt(Var(y) \times Var(y')))}$, where Cov denotes covariance, Var denotes variance.
>
> **MA** denotes multi-class accuracy, which is similar to BA. We treats the score ranging from -3 to 3 as seven classes (i.e., -3, -2, -1, 0, 1, 2, 3). MA is computed by the number of accurate-prediction samples dividing the total number.
>
> **MAE** denotes Mean-Absolute Error, which measures the L1 distance between prediction results and GT.
>
> On multimodal video retrieval, we follow MMT to use R@K and MdR.
>
> **R@K** denotes Recall@K. Taking the text-video retrieval as example, we first utilize the text retrieval K videos, if the GT video is in these K videos, we increase the quantity by one. The final result is calculated via dividing the total number.
>
> **MdR** denotes median rank. Taking the text-video retrieval as example, we first utilize the text to retrieval GT video and record the rank of GT video. We reorder the GT ranks of all the samples and calculate their median value, which is MdR.

---

> > ### Author Response · Authors · 2022-11-17
> > **The response for Reviewer SiTt (2)**
> >
> > **2.3.	About the normalization of dataset split**
> >
> > Thanks for the advice. We simple employ the L1 normalization, since L1 norm is more like a random scene in real life (than like exponential norm).
> >
> > **2.4.	About Even MulT utilizes Transformer**
> >
> > IPD is implemented based on LMF which utilizes LSTM for feature encoding. In general conditions, the performance with Transformer is better than that of LSTM. Thus, we run the experiments with Transformer as a comparison. The performance of IPD is better than the LSTM-based baselines and even the Transformer-based one, which further demonstrates the effectiveness of IPD.
> >
> > **2.5.	About most of the methods (MMIN, MCTN) can not be utilized for 7 modalities**
> >
> > The reviewer can refer to the papers of MMIN and MCTN, these methods are designed specifically for fixed three modalities with a poor generalization.

---

### Author Response · Authors · 2022-11-17
**Thank you all for your questions and feedback**

We would like to thank reviewers for providing high-quality reviews and constructive feedback that have improved the paper. We are encouraged that reviewers think our paper is “well motivated, described, and the idea is clear” (Reviewer pCS1), “the idea is interesting, the key points are easy to follow” (Reviewer QUyJ), “The method seems to be novel” (Reviewer SiTt).

In our previous experiences, constructing tensor-theory based paper/work is difficult to some extent, since the clarity of equations and figures  should be seriously considered. We have tried our best to present the work based the previous successful experiences, by giving the sizes of all the variables, adding the connection between equations (variables) and corresponding modules, providing the concrete implementation process. Therefore, we hope that the revised version can be satisfactory.

We have updated our draft to further improve the writing and incorporate suggestions (marked in the revision) from reviewers and extended the appendix with more details about experiments. **The explicit motivations of all the modules are red-color-coded, and the improvements are blue-color-coded (some improvements are in the appendix). Concretely,**

(1) We extend the related work with more methods and illustration.

(2) We polish Fig. 2 by connecting the equations, variables (vectors) with corresponding modules, and adding detailed illustration.

(3) We polish Fig. 3 by adding more details.

(4) We polish Eqs. 1 and 2 for more convenient understanding.

(5) We add the results of T2FN and random drop in Table 1 and 2.

(6) We add the trade-off results of performance and computation in Table 5.

(7) We add the experiments where all the modalities are available during training in Table 8 (Appendix).

(8) We add the experiments of other datasets in Table 9, 10, 11 (Appendix).

---

### Decision · Program_Chairs · 2023-01-20

**Decision:**

Reject

**Justification For Why Not Higher Score:**

I could not ascertain that the authors have done a decent enough job of adapting the baselines - in simple, common-sense ways - to their setting.

**Justification For Why Not Lower Score:**

N/A

**Metareview: Summary, Strengths And Weaknesses:**

The paper presents a new method for multimodal learning in the event of missing modalities, both during training time and at test time.
I am basing my meta-review mostly on the assessments by Reviewers SiTt and Reviewer pCS1, who seemed both knowledgeable on the topic and seemed to have taken the time to read the paper. The method appears to be sufficiently novel and it's designed for a relatively under-explored setting wherein a modality is missing (or partially missing).
Some issues regarding clarity were raised, with the authors having handled some of them (Fig 2), through more might be done, such as including the responses for 2.1,2.2 and 2.3 in the papers, as suggested by SiTt.
The main weakness of the paper is the experimental evaluation, which is tricky to do because the method is designed to work in a setting not addressed by previous methods. However, most prior work is not designed to work under these conditions, so a reasonable effort must be made by the authors to make sensible adaptations of existing methods to this setting, such as fine-tuning the baselines on whatever modalities exist during training, and/or pre-processing the data to perform some imputation of the missing values (even just mean imputation). Unfortunately, even though I read the experimental section, I could not discern whether this has been done, as details are missing.

Moreover, the authors might want to include the following contenders for MOSI/MOSEI in a future version:
MAGBERT/MAGXLNET (Rahman et al 2020)
DynMM (Xue and Marculescu, 2022)


**Summary Of Ac-Reviewer Meeting:**

N/A